# Detergent modulates the conformational equilibrium of SARS-CoV-2 Spike during cryo-EM structural determination

Shawn B. Egri[1,5], Xue Wang[2,5], Marco A. Díaz-Salinas [3,5], Jeremy Luban [1,4], Natalya V. Dudkina [2] ✉, James B. Munro [3] ✉ & Kuang Shen [1,4] ✉

The Spike glycoprotein of SARS-CoV-2 mediates viral entry into the host cell via the interaction between its receptor binding domain (RBD) and human angiotensin-converting enzyme 2 (ACE2). Spike RBD has been reported to adopt two primary conformations, a closed conformation in which the binding site is shielded and unable to interact with ACE2, and an open conformation that is capable of binding ACE2. Many structural studies have probed the conformational space of the homotrimeric Spike from SARS-CoV-2. However, how sample buffer conditions used during structural determination influence the Spike conformation is currently unclear. Here, we systematically explored the impact of commonly used detergents on the conformational space of Spike. We show that in the presence of detergent, the Spike glycoprotein predominantly occupies a closed conformational state during cryo-EM structural determination. However, in the absence of detergent, such conformational compaction was neither observed by cryo-EM, nor by single-molecule FRET designed to visualize the movement of RBD in solution in real-time. Our results highlight the highly sensitive nature of the Spike conformational space to buffer composition during cryo-EM structural determination, and emphasize the importance of orthogonal biophysical approaches to validate the structural models obtained.

In early December 2019, the emergence of a deadly, zoonotic coronavirus, severe acute respiratory syndrome coronavirus 2 (SARS-CoV-2), occurred in the Hubei province of central China[1]. SARS-CoV-2 is a single-stranded RNA-enveloped virus belonging to the β coronavirus family[2]. The viral capsid is composed of four structural proteins—S, E, M, and N—which enclose the RNA genome of the virus[3]. Glycosylated S (Spike) proteins cover the surface of the virion and bind the host cell receptor angiotensin-converting enzyme 2 (ACE2) in order to mediate viral entry[4]. Upon entry, the viral RNA genome is released, which then

undergoes translation, replication, and transcription to promote the synthesis of new viral particles within the host cell[5].

The homotrimeric Spike protein resides on the viral surface and is the basic unit responsible for facilitating host receptor recognition, cell attachment, and viral fusion during infection[6–8]. In the absence of host cell interaction, the Spike protein exists in a stable, prefusion state which undergoes major structural rearrangements upon interaction with the host ACE2 receptor[9,10]. SARS-CoV-2 Spike consists of an extracellular N-terminus, a transmembrane domain anchored within

[1]Program in Molecular Medicine and Department of Biochemistry & Molecular Biotechnology, University of Massachusetts Chan Medical School, 373 Plantation St, Worcester, MA 01605, USA. [2]Thermo Fisher Scientific, Achtseweg Noord 5, 5651 GG Eindhoven, The Netherlands. [3]Department of Microbiology and Physiological Systems, University of Massachusetts Chan Medical School, 364 Plantation St, Worcester, MA 01605, USA. [4]Massachusetts Consortium on Pathogen Readiness, Boston, MA, USA. [5]These authors contributed equally: Shawn B. Egri, Xue Wang, Marco A. Díaz-Salinas. ✉e-mail: natalya.dudkina@thermofisher.com; James.Munro@umassmed.edu; Kuang.Shen@umassmed.edu

the viral membrane, and a small intracellular C-terminal domain[11]. Within the extracellular domain, each Spike protomer contains a receptor binding domain (RBD) which can adopt one of two states—a closed, or open state[6,12,13]. When closed, the ACE2-binding interface faces downwards and is shielded, making it unavailable for binding. Upon transitioning to the open state, the RBD ACE2-binding site points upwards and becomes exposed, making it available for binding[14,15]. This switch between up and down conformations suggests that Spike RBDs must transition to the open state prior to engaging the ACE2 receptor. Discerning how the global conformational state of the Spike protein impacts ACE2 binding is necessary for furthering our understanding of pathogenesis and aiding the development of therapeutics that disrupt ACE2 binding and subsequent membrane fusion.

As SARS-CoV-2 Variants continue to emerge bearing complex combinations of non-synonymous Spike mutations associated with heightened transmissibility, it is important to determine how these mutations impact the conformational state of the protein[16,17]. As the Spike RBD plays a central role in ACE2 recognition and binding, it is predicted that changes to the amino acid sequence of the RBD impact the molecular mechanism of the interaction between Spike and ACE2, resulting in changes to the viral infectivity. This is highlighted by the fact that most mutations in the SARS-CoV-2 variants of concern occur on the surface of the Spike protein, and specifically many occur within the RBD[18]. Numerous studies aimed at describing how mutations to the Spike protein are correlated with heightened infectivity have employed cryogenic electron microscopy (cryo-EM) to explore the global conformational space of Spike Variants (Supplementary Table 1)[9,13,19,20]. In order to accurately determine how Spike mutations impact the conformational dynamics of the protein, we need a precise understanding of the biophysical tools and methods commonly employed to investigate the SARS-CoV-2 Spike protein.

In this study, we explore the structural impact of sample buffer composition on the Spike protein conformation. We found that the addition of commonly used detergents, CHAPS or DDM, induces a transition of the Spike RBD from an open to a more closed conformation during cryo-EM structural determination at either physiological pH or at acidic pH. Importantly, when we extended our analysis to single-molecule FRET (smFRET), we found that the presence of detergent did not influence RBD conformation, which is in sharp contrast to the structural determination by cryo-EM. Our results reveal that the Spike RBD conformation is highly sensitive to its environment during cryo-EM structural determination and emphasize the importance of using orthogonal biophysical approaches when investigating conformational equilibrium and dynamics of the SARS-CoV-2 Spike protein.

## Results

### Structural determination of the D614G SARS-CoV-2 Spike in the absence of detergent

To investigate the conformational space of SARS-CoV-2 Spike protein, we used the D614G variant as a model system as it was the first mutation to increase infectivity and, as seen below, it expands the conformational space relative to the original Wuhan-Hu-1 Spike[13,21–23]. We purified soluble homotrimeric Spike protein using a previously defined approach[13]. Briefly, His-tagged Spike was recombinantly overexpressed in HEK293-F cells and secreted to the media. We then pulled down the Spike protein using Ni-NTA resin, and further purified it using size-exclusion chromatography (Supplementary Fig. 1a). This approach yielded high-quality SARS-CoV-2 Spike in its intact, homotrimeric form, which is suitable for cryo-EM structural determination (Supplementary Fig. 1b, c).

To set up the benchmark, we carried out cryo-EM single particle analysis on the purified Spike protein without detergent (Table 1 and Supplementary Fig. 1d). Well-defined particles were clearly seen in the micrographs taken by a 300 kV cryo-electron microscope

(Supplementary Fig. 1e). Reference-free two-dimensional (2D) classification revealed identifiable structural details of the homotrimeric protein complex (Supplementary Fig. 1f). Masked three-dimensional classification and refinement identified four distinct populations of particles (Fig. 1a, b): 5% of particles adopted a fully closed conformation (4.3 Å, gold-standard criteria), with all three RBDs facing downwards, 36% of particles had one RBD standing up (3.9 Å, gold-standard criteria), 40% of particles had two RBDs up (4.0 Å, gold-standard criteria), and 19% of particles were in the fully open, three-RBD-up conformation (3.9 Å, gold-standard criteria). The general features of these four maps are in agreement with those that have been previously published, with similar proportions of particles found between the four classes[13]. We rigid-body docked previously generated domains of the open and closed Spike protein, and further refined them based on the corresponding map, to generate four distinct atomic structural models (Fig. 1a).

When we compared our D614G Spike results to other studies we found apparent differences. Common throughout all previously published studies is the observation that the D614G mutation disrupted a critical interprotomer hydrogen bond, located between adjacent subunits[13,24,25]. However, distinct from previous studies is the reported impact of the mutation on the global conformation (i.e., "openness") of the Spike protein. Here, we observed a variety of conformations, with the three RBDs within the homotrimer occupying 1-up, 2-ups, and 3-ups conformation, while other published studies reported a more closed global conformation (Fig. 1b)[24,25]. This difference raised discrepancies regarding data interpretation and mechanism of how D614G acquired higher virulency and out-competed the original D614 strain.

### Impact of detergent on cryo-EM structural models of the SARS-CoV-2 Spike at physiological pH

We carefully examined the conditions in which different structural models were resolved, and noticed a major difference between these studies - the use of detergents in sample buffers during cryo-EM sample preparation. As detergents are usually chosen heuristically to reduce the preferred orientation of particles in vitrified ice on cryo grids, we suspected they may be a key factor to the observed discrepancy. Accordingly, we systematically characterized how detergents impact the global conformation of the Spike protein. We tested two commonly used detergents, CHAPS and DDM. CHAPS (3-((3-cholamidopropyl) dimethylammonio)−1-propanesulfonate) is a zwitterionic detergent, while DDM (n-dodecyl-D-maltoside) is nonionic. Both detergents have been widely used in cryo-EM sample preparation[26,27]. Here, following Spike protein purification and immediately preceding cryo-EM grid plunging, we added a specific detergent to the sample. Grids were then prepared to carry out single particle analysis. In all the conditions tested, we clearly resolved particles (Supplementary Figs. 2–5). Further 2D classification revealed identifiable structural details of the homotrimeric protein complex that allows for ab initio structural model generation and classification of different conformations (Supplementary Figs. 2–5).

For the sample containing 0.01% CHAPS, we obtained four distinct populations of particles following three-dimensional classification and refinement, which were comparable in structural details and resolution to the classes identified in the detergent-free dataset (Fig. 2a, 3.9 Å, 3.7 Å, 3.8 Å, and 4 Å resolution for all-closed, 1-up, 2-ups, and all-open conformation, respectively). However, in contrast to the detergent-free samples, the distribution of Spike conformations shifted towards a more closed state (Fig. 2b). Specifically, 10% of particles displayed a fully closed conformation with all RBD's in the down conformation, 42% of particles displayed the one-RBD-up conformation, 37% of particles showed two RBD's in the up conformation, and 11% of particles were in the fully open conformation (Fig. 2b). We observed a more striking shift when we increased the CHAPS concentration to 0.5%

**Table 1 | Summary of cryo-EM data collection conditions**

| Data collection and processing | | | | | | | | | |
|---|---|---|---|---|---|---|---|---|---|
| Spike | D614G | D614G | D614G | D614G | D614G | D614G | D614 | D614G | D614 |
| pH | 7.4 | 7.4 | 7.4 | 7.4 | 7.4 | 5 | 5 | 5 | 5 |
| Additive | No detergent | 0.01% CHAPS | 0.5% CHAPS | 0.01% DDM | 0.5% DDM | No detergent | No detergent | 0.5% CHAPS | 0.5% CHAPS |
| Microscope | Thermo Scientific Krios G4 | | | | | | | | |
| Magnification | 76,000 | | | | | | | | |
| Voltage (kV) | 300 | | | | | | | | |
| Grids | UltrAuFoil R1.2/1.3 300 mesh or UltrAuFoil R0.6/1 300 mesh | | | | | | | | |
| Electron exposure (e⁻/Å²) $\,(e^-/Å^2)$ | 37 | 40.54 | 41.17 | 42 | 41 | 42 | 39.16 | 40 | 40 |
| Defocus range (μm) | –0.6 to –1.8 | –0.7 to –1.3 | –0.7 to –1.3 | –0.7 to –1.3 | –0.7 to –1.3 | –0.7 to –1.3 | –0.7 to –1.3 | –0.7 to –1.3 | –0.7 to –1.3 |
| Pixel size (Å) | 1.053 | 1.053 | 1.053 | 1.053 | 1.053 | 1.053 | 1.053 | 1.053 | 1.053 |
| Symmetry imposed | C3 | | | | | | | | |
| Final number of particles | 266296 | 294252 | 260885 | 399407 | 193543 | 637756 | 687148 | 421644 | 218968 |
| Map resolution | See Supplementary Table 2 | | | | | | | | |
| FSC threshold | 0.143 | | | | | | | | |
| Map resolution range | See Supplementary Table 2 | | | | | | | | |

(Supplementary Fig. 3): 16% of particles in the fully closed conformation, 49% in the one-RBD open position, 30% in the two-RBD open conformation, and 5% displaying the fully open conformation (Fig. 2b). We also observed that at higher CHAPS concentration the particles no longer displayed a strong preference towards a single orientation (Supplementary Fig. 3e, cf. Supplementary Fig. 1h). Taken together, these results suggest that addition of a commonly utilized detergent, CHAPS, induces the Spike protein to adopt a more closed conformation in cryo-EM structural analysis.

Next, we asked if the CHAPS-induced Spike RBD closing extends to DDM. Following protein purification, we added DDM (0.01% or 0.5%) to the sample immediately prior to grid plunging, and a similar streamline of data analysis was performed (Supplementary Figs. 4 and 5). We again observed the SARS-CoV-2 Spike protein in a more closed conformation relative to the detergent-free condition (Fig. 2b). With 0.01% DDM, 29% of particles displayed the fully closed conformation, 56% displayed the one open conformation, and 15% displayed the two-open conformation, with no particles observed in the fully open state. Furthermore, when 0.5% DDM was doped-in, 18% of particles were fully closed, 51% were in the one-RBD open state, 27% were in the two-RBD open state, and only 4% displayed the fully open conformation. We again observed that at the higher concentration of DDM the particles no longer displayed a strong preference for a single orientation (Supplementary Fig. 5e, cf. Supplementary Fig. 1h). Taken together, this result reinforces the notion that detergent promotes the closing of Spike RBDs.

### Detergent promotes conformational compaction at low pH

To further explore the impact of sample conditions on the SARS-CoV-2 Spike protein, we next monitored the RBD conformation at acidic pH. During protein purification, the sample buffer was adjusted to pH 5 by 50 mM sodium acetate during size-exclusion chromatography (Fig. 3a). We did not observe any changes in elution profiles for either the D614 or D614G Spikes, which ensures that lowering the pH does not impact the general organization of the Spike homotrimer (e.g., complex disassembly, or protomer unfolding). As a benchmark, we first analyzed the conformational space of the D614G SARS-CoV-2 Spike protein at pH 5. When subjected to cryo-EM analysis (Supplementary Fig. 6), we observed that acidic pH promotes a more open conformation. Following three-dimensional classification and refinement, we observed ten distinct particle classes (Fig. 3b). These ten-classes include the previously seen four classes (fully closed, one-, two-, or three-RBD open) as well as six additional classes, in which the

density of at least one up-RBD is scattered due to greater flexibility. For example, the fully open, three-RBD-up conformation is represented by four distinct classes: the first class where the RBD densities for all three subunits are observed, the second class where one-RBD density is scattered, and the third, and fourth classes display scattered densities for either two, or three RBD's, respectively.

When we carried out the same analysis on the ancestral SARS-CoV-2 D614 Spike protein (Supplementary Fig. 7), we observed a similar trend—acidic pH promotes a more open conformation (Fig. 3c). When subjected to cryo-EM analysis, we observed a distribution of all-closed (9%), one- (32%), two- (36%), and three-RBD open (23%) states (Fig. 3d). This result contrasts the situation at physiological pH (pH 7.4), as it has previously been shown that the ancestral D614 Spike protein only resides in the fully closed (47%), or one-RBD open (53%) states (Fig. 3d)[6]. Taken together, these results suggest that acidic pH promotes a more open Spike conformation and that the Spike RBD conformation is highly sensitive to its environment.

Based on the results in the previous section, we sought to ask how the combined addition of detergent (0.5% CHAPS) and acidic pH (pH 5) alters the conformation of the Spike (Supplementary Fig. 8 and Supplementary Fig. 9). Strikingly, when we analyzed both the ancestral D614 Spike protein as well as the D614G mutant, we found that 100% of particles for both datasets displayed the fully closed conformation at pH 5 in the presence of CHAPS (Fig. 3b). The resolution of the maps is at 2.7–2.8 Å, revealing fine details of the Spike protein at near-atomic resolution (Supplementary Figs. 8d and 9d). This result indicates that the detergent-induced Spike RBD closing is accentuated by acidic pH.

### Visualization of Spike protein conformational space by single-molecule FRET

Summarizing the results above, we were surprised by the ability of commonly used detergents to modulate the conformational space of Spike protein during cryo-EM structural determination. However, whether these observations reflect the intrinsic biophysical properties of Spike, or mere artifacts in a specific laboratory setting, is unknown. Accordingly, an independent biophysical approach to test the effect of detergents is absolutely necessary.

We decided to employ single-molecule FRET (smFRET) as an orthogonal approach to independently explore the conformational space of SARS-CoV-2 Spike[15,28]. We site-specifically attached LD550 and LD650 fluorophores to the RBD and NTD domains of the Spike protein, respectively (Fig. 4a, see "Methods" for details). When the RBD is in the closed conformation, a high (~0.65) FRET state is observed. When the

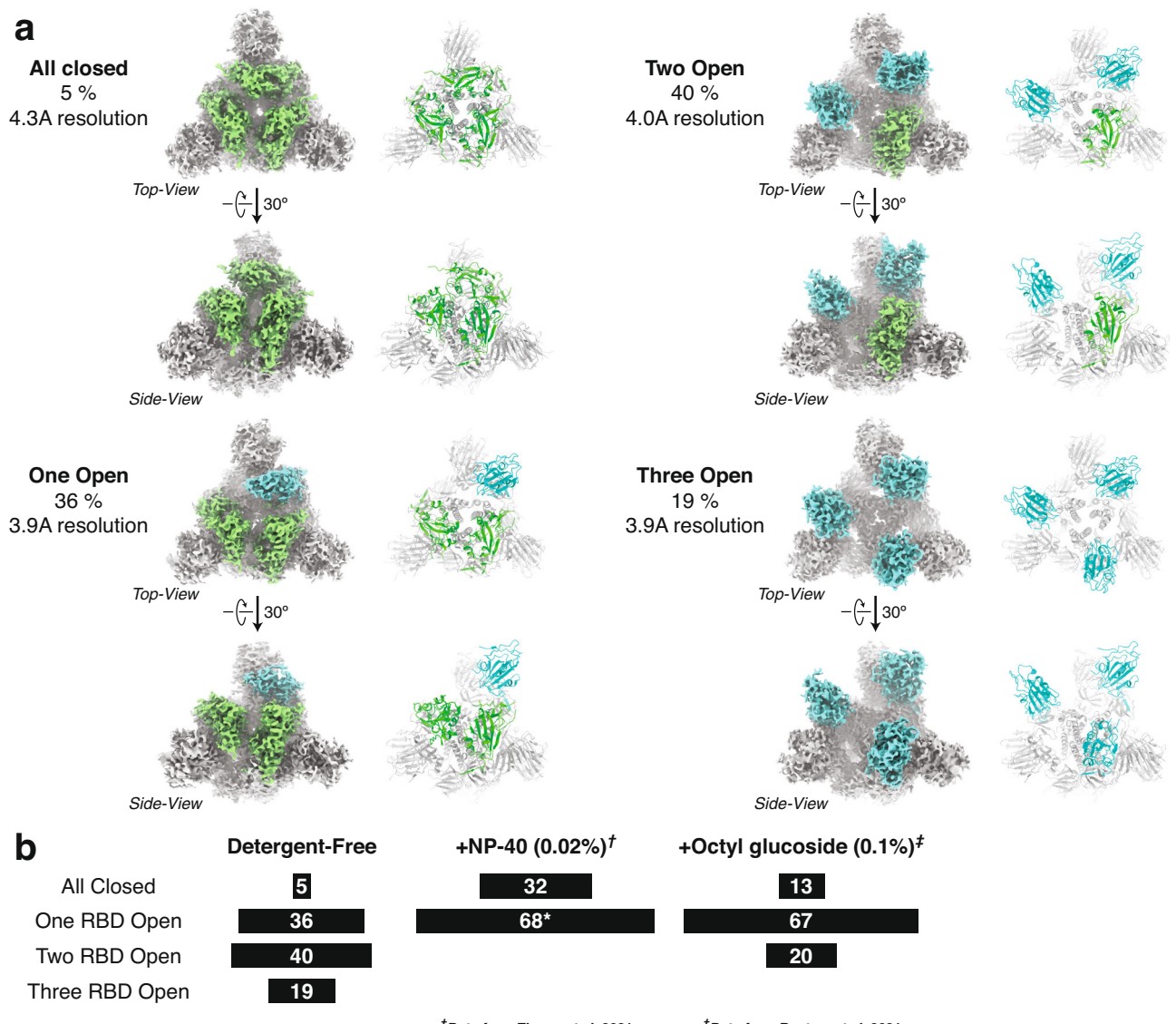

**Fig. 1 | SARS-CoV-2 D614G Spike protein displays open conformation in the absence of detergent. a** Cryo-EM density maps and structural models for the four classes of SARS-CoV-2 D614G Spike conformations—fully closed (upper left), one RBD (bottom left), two-RBD open (top right), and fully open (bottom right). **b** Summary of distributions of SARS-CoV-2 D614G Spike conformations at physiological pH. Dagger denotes data which was sourced from Zhang et al.[24]. Double dagger denotes data which was sourced from ref. 25. Asterisk denotes that the one RBD open and one-RBD partial-open states identified in Zhang et al. were combined for simplicity.

RBD transitions to the open conformation, the distance between the fluorophores increases, leading to a low (~0.35) FRET state. Therefore, this assay allows for monitoring the conformation of Spike RBD in real time. We first tested the conformational space of D614 and D614G Spike proteins in the absence of detergents. We observed frequent transitions between the high and low-FRET states in both cases, consistent with the dynamic nature of the Spike RBD (Fig. 4b, d). As previously reported, we observed that D614G Spike increases the population of an open RBD from 40 to 60% in comparison to the D614 Spike (Fig. 4c, e, no detergent)[28]. This result is also consistent with our cryo-EM structural analysis in the absence of detergents, showing that the D614G mutation promotes the RBD-up conformation in the Spike protein.

We next sought to determine if the presence of detergents, CHAPS and DDM, impacts the openness of the Spike protein RBD. Surprisingly, we observed that the presence of CHAPS or DDM does not shift the occupancy of the D614G Spike protein RBD towards a more closed state (Fig. 4c). In contrast, the FRET distribution of the high- and low-FRET states are almost identical to the case without detergents (Fig. 4c). This result sharply contradicts the cryo-EM structural analysis, and suggests that CHAPS and DDM do not impact the openness of the SARS-CoV-2 Spike protein in solution in our smFRET assay.

Finally, we asked if the conformational compaction of Spike protein by detergent at acidic pH can be validated using smFRET. We first measured smFRET distribution of D614 and D614G Spike at acidic pH without detergent. Similar FRET transitions between the high- and low-FRET states were observed, corresponding to the closed- and open conformation of RBD (Supplementary Fig. 10). At pH 5, the low-FRET state is more populated in comparison to pH 7.4, suggesting both D614 and D614G Spikes adopt a more open conformation (Fig. 4c, e). This is consistent with the cryo-EM structural analysis in the absence of detergent, in which both D614 and D614G Spike proteins showed a greater preference towards the RBD open position as compared to at pH 7.4. Importantly, the addition of detergent (0.5% CHAPS) does not alter the conformational distribution of the Spike RBDs in our single-

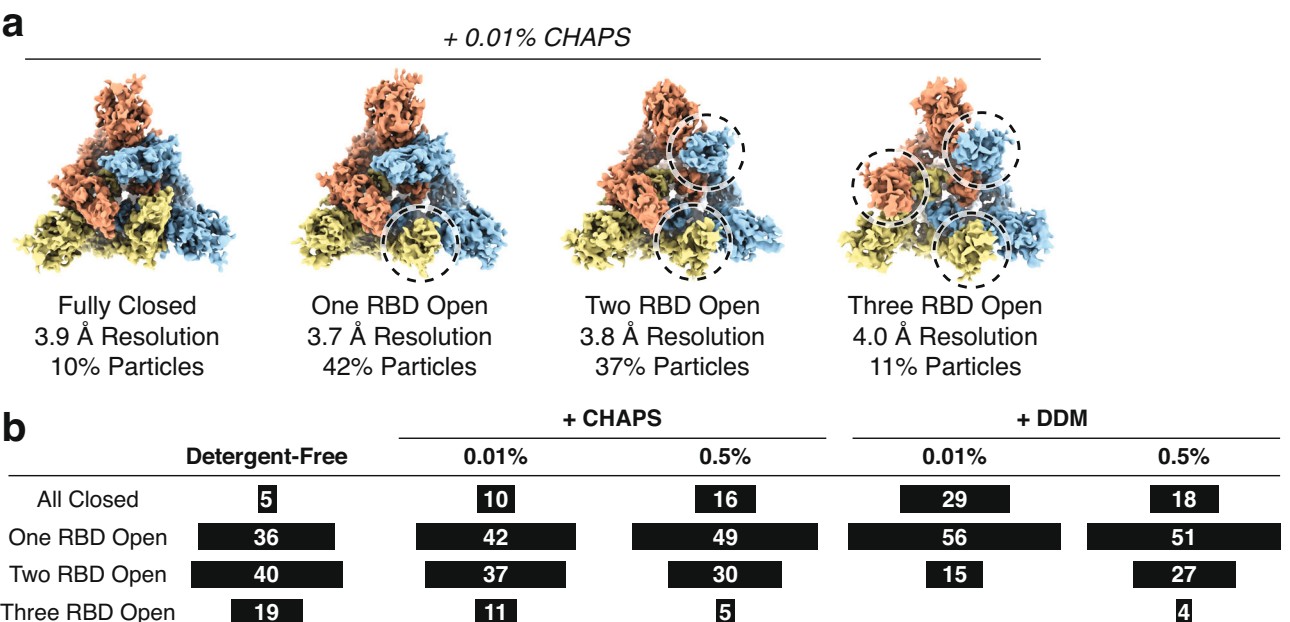

**Fig. 2 | Detergent shifts conformation of SARS-CoV-2 Spike protein toward a more closed state. a** Cryo-EM density maps for the four classes of SARS-CoV-2 D614G Spike conformations in the presence of 0.01% CHAPS. **b** Summary of distributions of SARS-CoV-2 D614G Spike conformations in the presence of CHAPS or DDM at various concentrations.

molecule assay—CHAPS does not compact the Spike conformation as it does in cryo-EM structural determination (Fig. 4c, e). Taken together, these results suggest that acidic pH promotes a more open conformation of the Spike protein, while the presence of detergent does not impact the RBD conformation in solution via smFRET.

## Discussion

In order to understand how the emergence of novel SARS-CoV-2 Variants correlates with enhanced viral infectivity, numerous studies have employed structural techniques such as cryo-EM to understand how mutations change the global conformational space of the Spike protein (Supplementary Table 1)[20]. These studies have been used to correlate Spike conformation of a particular strain with their biological properties, including transmissibility, infectivity, and immune escape. In addition, they have aided our attempts to design more efficacious therapeutics through structure-based antibody design[29,30]. Comparisons across these studies are frequently made, but experimental conditions are often overlooked when making these comparisons. To reach biologically-relevant conclusions and to ensure consistency across these structural studies, it is necessary to confirm that structural data accurately depict the conformational state of the Spike protein. To date, no study has probed how cryo-EM sample buffer conditions impact the conformation of the SARS-CoV-2 Spike protein.

In this study, we systematically analyzed how the addition of commonly used detergents in sample preparation buffers impact the global conformational space of the SARS-CoV-2 Spike protein. Using the D614G Spike protein variant as a model system, we first show that in the absence of commonly used detergents, the D614G Spike protein displays a variety of conformations (Fig. 1), and acidic pH promotes a more open conformation (Fig. 3). In contrast, upon addition of CHAPS or DDM to sample preparation buffers, the distribution of Spike conformations shifts to a more closed state either at physiological pH or acidic pH (Figs. 2 and 3 and Supplementary Table 2). Lastly, we found that the detergent-induced closing of Spike RBDs during cryo-EM structural determination does not extend to single-molecule FRET measurements (Fig. 4). These results suggest that the Spike RBD is sensitive to detergent during structural determination with cryo-EM, but not with smFRET.

Detergents are commonly employed during structural determination to improve the quality of the cryo-EM grids, which occurs by (i) increasing grid surface hydrophilicity, and (ii) reducing the preference of particles to coalesce with the air-water interface[31]. The use of detergents on glow-discharged grids increases grid surface hydrophilicity[27], which reduces surface tension by strengthening the interaction between the water molecules in the buffer and the surface of the grid. Furthermore, the addition of detergent reduces the tendency of proteins to adhere to the air-water interface within vitrified ice, as this phenomenon commonly results in protein denaturation as well as a preferred orientation of the protein particles, which could lead to a poor 3D reconstruction[32]. Although in this study we use the SARS-CoV-2 Spike protein as a model system to address how detergents change the protein conformation during cryo-EM structural determination, these observations may be broadly applicable to other protein structures resolved by cryo-EM as detergents are often included in the sample buffer. Our results raise a concern that direct comparison of protein conformations determined under different buffer compositions should be evaluated carefully, as detergents may be responsible for introducing undesirable artifacts.

In our cryo-EM data, we observe that in detergent-free conditions the particles display a strong preference towards a single orientation (Supplementary Fig. 1h). As such, we collected the detergent-free dataset with a tilt-angle of 30-degrees in order to improve 3D reconstruction (see Methods & Materials for details). Upon addition of a high concentration of detergent, we observe a more uniform distribution of angular orientations, indicating that the particles no longer display a preference for a single orientation (Supplementary Figs. 3e and 5e; cf. Supplementary Fig. 1h). Although initially this appears promising as detergents resolve the preferred-orientation issue common throughout cryo-EM structural studies, it is ultimately undesirable as it is also responsible for altering the conformation of the Spike (i.e., RBD closing) which significantly changes the biological interpretation (Fig. 2 and Supplementary Tables 1 and 2). While it has been previously shown that the addition of detergents resolves the preferred orientation issue by shifting the physical location of particles from the air-water interface to being embedded in the

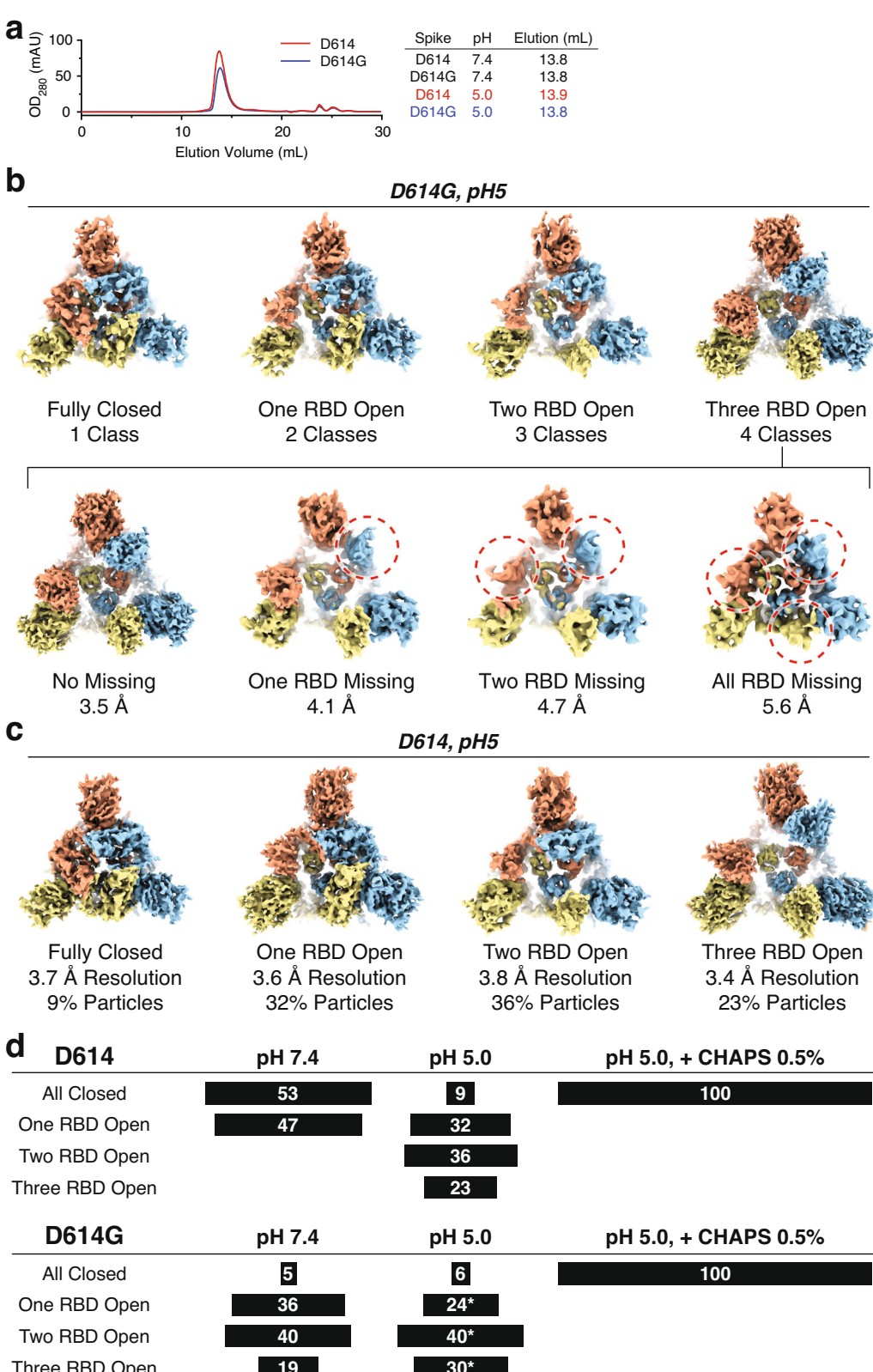

**Fig. 3 | Acidic pH promotes the opening of SARS-CoV-2 Spike protein conformation. a** Gel-filtration profiles for the SARS-CoV-2 D614 and D614G Spike proteins at acidic pH (pH 5.0). **b** Cryo-EM density maps for the SARS-CoV-2 D614G Spike protein at acidic pH (pH 5) represented by ten different classes. Four of the ten classes display fully resolved RBDs (upper); the remaining six classes display scattered density for at least one RBD (lower). Red dashed circles represent scattered RBD density. **c** Cryo-EM density maps for the four classes of SARS-CoV-2 D614 Spike conformations at acidic pH (pH 5). **d** Summary of distributions of SARS-CoV-2 D614 and D614G Spike conformations at acidic pH (pH 5) in the absence or presence of 0.5% CHAPS. Asterisk denotes that the SARS-CoV-2 D614G Spike conformations with scattered RBD densities were combined with their respective one-, two-, or three-RBD-up classes.

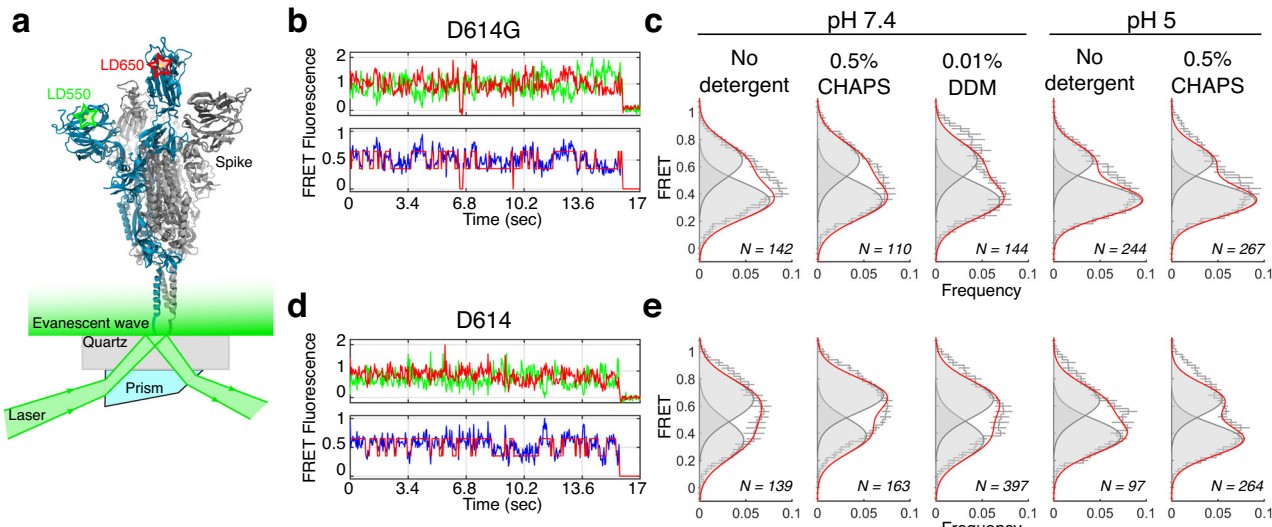

**Fig. 4 | Detergent-induced opening of SARS-CoV-2 Spike protein does not extend to single-molecule FRET measurements. a** Schematic representation of single-molecule FRET setup. Approximate locations of fluorophores represented by green (LD550) and red (LD650) stars on the N-terminal domain (NTD) and receptor binding domain (RBD), respectively. **b, d** Representative single-molecule FRET trace acquired for the SARS-CoV-2 D614G Spike (**b**) and SARS-CoV-2 D614 Spike (**d**). Idealization resulting from Hidden Markov modeling (HMM) analysis is overlaid (red). The high-FRET (0.6) and low-FRET (0.4) states correspond to the RBD-down and RBD-up conformations, respectively. **c, e** FRET histogram for SARS-CoV-2 D614G Spike (**c**) and SARS-CoV-2 D614 Spike (**e**) with the sum of two Gaussian distributions centered at 0.6 and 0.4 FRET (sum, red; single distributions, gray) generated from the results of HMM analysis. FRET histograms are presented as mean ± standard error determined from three replicates. The total number of smFRET traces used in the HMM analysis is indicated. Source data are provided as a Source Data file.

vitreous ice, we speculate that this change in physical environment may also be responsible for altering the Spike conformation[32]. Whether the physical environment encountered at air-water interfaces has a role in the transmission of respiratory viruses such as SARS-CoV-2 warrants future investigation.

Lastly, we show that the detergent-induced RBD closing observed during cryo-EM structural determination is not observed in solution when evaluated with smFRET. Spike RBD displays similar degrees of openness independent of the presence of detergent—approximately 40% RBD-up for D614 and approximately 60% RBD-up for D614G, in either the presence or absence of CHAPS or DDM, at physiological pH (Fig. 4). In this setup, the protein resides within an aqueous environment instead of a thin layer of vitrified ice, therefore the addition of detergent does not redistribute or reorient the Spike molecules and thus has no impact on the openness of the RBD. Furthermore, this result is in alignment with our previous cryo-EM structural characterization of the D614G Spike global conformation in the absence of detergents, namely an RBD opening induced by this mutation[13]. Despite differences in experimental design between cryo-EM and smFRET (e.g., Spike construct, fluorophore attachment, temperature differences), reaching a similar result via an orthogonal approach reveals that using multiple biophysical techniques to investigate conformational dynamics may be a useful approach to understanding SARS-CoV-2 Spike protein dynamics. In other fields of study, cryo-EM and smFRET have been used in tandem, and there are now numerous studies showing that these techniques can be used synergistically[33]. In conclusion, we show that the conformational equilibrium and dynamics of SARS-CoV-2 Spike RBD is sensitive to its environment, and highlight that the presence of commonly used detergents must be considered when interpreting cryo-EM structural determination of the SARS-CoV-2 Spike protein.

## Methods
### Plasmids
The mammalian codon-optimized gene coding soluble His-tagged SARS-CoV-2 Spike ectodomain from Wuhan-Hu-1 strain and its D614G variant with stabilizing 2Pro and Furin cleavage site mutations are deposited in Addgene (D614: Addgene plasmid # 164652; D614G: Addgene plasmid # 164651) and have been described previously[11].

The mammalian codon-optimized gene coding SARS-CoV-2 Spike ectodomain (SΔTM) from Wuhan-Hu-1 strain (D614, GenBank ID: MN908947.3), its D614G variant, and their versions with A4-peptide (DSLDMLEW) insertions at amino acid positions 161 and 345 have been described before[13,28].

### Purification of soluble SARS-CoV-2 Spike protein homotrimers
The purification of SARS-CoV-2 Spike homotrimers was carried out using a previously developed protocol[13]. Briefly, FreeStyle 293-F cells (ThermoFisher R79007) were cultured in SMM-293 TII serum-free media (SinoBiological) in a 37 °C shaker at 8% $CO_2$ and 80% humidity. 400 µg of plasmid encoding His-tagged SARS-CoV-2 Spike protein was transfected into 400 mL of 293 FreeStyle cells at $5 \times 10^5$ cells/mL, using 30 mL Opti-MEM and 1.2 mL PEI. 72 h later, the media was collected, filtered (0.45 µm), and then incubated with 5 mL of Ni-NTA resin (Qiagen) for 60 min at 4 °C. The resin was collected and washed with PBS supplemented with 15 mM imidazole to mitigate non-specific proteins. Soluble Spike protein was eluted using PBS supplemented with 200 mM imidazole. Following, the Spike protein was further enriched using a Superose 6 gel-filtration column (Cytiva), in a buffer containing 25 mM NaHEPES (pH 7.4) and 150 mM NaCl.

### Screening of detergent conditions for cryo-EM structure determination
To screen various commonly used detergents during cryo-EM data collection, the Thermo Scientific™ VitroEase™ Buffer Screening Kit was used. The cryo-EM sample solutions with a final concentration of 0.5% (w/v) detergents were prepared via mixing 9 µL - 6.5 mg/mL spikes and 1 µL one of the following detergents (CTAB, CHAPS, OG, Tween-20, or FOM) from the kit. DDM was also tested. For each sample, 3 µL sample was deposited on an UltrAuFoil R1.2/1.3 300 mesh grid that had been glow-discharged for 30 s in a GloQube Plus Glow Discharge System. Plunge freezing was performed with a Vitrobot Mark IV using a blot

force 0 and blot time 5 s at 100% humidity and 4 °C. All frozen grids were imaged with a Thermo Scientific Krios G4 Cryo-Transmission Electron Microscope (Cryo-TEM) operated at a fixed 300 kV, and images were recorded with a Falcon4 Direct Electron Detector in EER format. Movies were acquired using EPU Multigrid software in one day. Approximately 500 micrographs were automatically collected for each sample with a defocus range between 0.6 and 1.1 μm, a pixel size of 1.05 Å/pixel and a total dose ~40 e/Å². The micrographs were processed with cryoSPARC Live during data collection[34].

## Cryo-EM sample preparation and data collection
Purified Spike was concentrated to ~6.5 mg/mL and detergents, CHAPS or DDM, were spiked-in to a final concentration of 0.01% or 0.5% (w/v). For the detergent-free condition, purified Spike was concentrated to ~3 mg/mL. For each sample, 3 μL sample was deposited on an UltrAuFoil grid that had been glow-discharged for 30 s in a GloQube Plus Glow Discharge System. Plunge freezing was performed with a Vitrobot Mark IV using a blot force 0 and blot time 5 s at 100% humidity and 4 °C. Detergent-free samples were plunge-frozen using UltrAuFoil R1.2/1.3 300 mesh grids, and detergent containing samples were plunge-frozen using UltrAuFoil R0.6/1 300 mesh grids. Grids were imaged with a Thermo Scientific Krios G4 Cryo-Transmission Electron Microscope (Cryo-TEM) operated at a fixed 300 kV, and images were recorded with a Falcon4 Direct Electron Detector in EER format. Movies were acquired using EPU 2 software. Micrographs were automatically collected for each sample with a defocus range between 0.7 and 1.3 μm, a pixel size of 1.05 Å/pixel and a total dose ~40 e/Å². Detergent-free samples were collected with a tilt-angle of 30° to overcome the preferred orientation.

## Cryo-EM data processing
Pre-processing (motion correction, estimation of microscope contrast-transfer function parameters, particle picking, and initial 2D classification) were carried out in cryoSPARC Live during data collection. Further 2D classifications and homogenous refinements using the published structure EMD-21452 as initial model were performed in cryoSPARC3.1. Particle stacks with well-refined orientation parameters were imported in Relion3.1. Focused 3D classifications with a soft mask on the S1 subunit of the protomer were performed on the C3 symmetry-expanded particles to identify the open and closed monomer conformations. Trimer classes were identified by 3D classification on the trimer particles, as was performed similarly in a previous study[13]. Particle stacks from each trimer class were imported back to cryoSPARC3.1 to perform homogeneous refinement.

## Preparation of samples for smFRET experiments
Expression, purification, and fluorescent labeling of SΔTM trimers for smFRET experiments have been reported previously[28]. Briefly, SΔTM hetero-trimers were expressed by co-transfection of ExpiCHO-S cells (ThermoFisher A29127, Waltham, MA, USA) with both the untagged SΔTM construct (D614 or D614G) and the corresponding 161/345 A4 peptide tagged SΔTM plasmid at a 2:1 molar ratio. After their purification by affinity chromatography using Ni-NTA agarose beads (Invitrogen™, Waltham, MA, USA) and size-exclusion chromatography, SΔTM hetero-trimers were labeled by overnight incubation at room temperature with coenzyme A (CoA)-conjugated LD550 and LD650 fluorophores (Lumidyne Technologies, New York, NY, USA) and Acyl carrier protein synthase (AcpS). SΔTM was purified away from unbound dye and AcpS by a second round of size-exclusion chromatography. Aliquots were stored at −80 °C until use. Expression and preparation of soluble human monomeric ACE2 (hACE2) for smFRET experiments have been described previously[28]. Briefly, pcDNA3.1(−) (Ampicillin^R) encoding human ACE2 ectodomain (residues 1–615) with a C-terminal His-tag was transfected in ExpiCHO-S cells, and the expression of ACE2 was driven by the CMV promoter in the pcDNA3.1(−) vector. Six days post-transfection, the supernatant containing ACE2 was harvested and the cell pellets were discarded. Supernatant buffer composition was adjusted to 20 mM Tris-HCl (pH 8.0), 20 mM imidazole, 500 mM NaCl, and 10% glycerol, and ACE2 was pulled down using Ni-NTA resin (Invitrogen, Waltham, MA, USA). After extensive wash in a buffer containing 20 mM Tris-HCl (pH 8.0), 20 mM imidazole, 500 mM NaCl, and 10% glycerol, ACE2 was eluted by a buffer containing 20 mM Tris-HCl (pH 8.0), 300 mM imidazole, 500 mM NaCl, and 10% glycerol. ACE2 was further purified by size-exclusion chromatography on a Superdex 200 Increase 10/300 GL column (GE Healthcare, Chicago, IL), in PBS.

## smFRET imaging
Labeled SΔTM spikes were immobilized on streptavidin-coated quartz microscope slides by way of Ni-NTA-biotin (Sigma-Aldrich, St. Louis, MO, USA) and imaged using wide-field prism-based TIRF microscopy that centers on a Rapid Automated Modular Mounting (RAMM) microscope frame[28,35–38]. Imaging was performed in either physiological (PBS, pH 7.4; Fisher Scientific, Hampton, NH, USA) or acidic (100 mM acetate buffer pH 5.0, 153 mM NaCl) conditions, as indicated. Imaging buffers also contained 1 mM Trolox (Sigma-Aldrich, St. Louis, MO, USA), 1 mM cyclooctatetraene (COT; Sigma-Aldrich, St. Louis, MO, USA), 1 mM 4-nitrobenzyl alcohol (NBA; Sigma-Aldrich, St. Louis, MO, USA), 2 mM protocatechuic acid (PCA; Sigma-Aldrich, St. Louis, MO, USA), and 8 nM protocatechuate 3,4-deoxygenase (PCD; Sigma-Aldrich, St. Louis, MO, USA) to stabilize fluorescence and remove molecular oxygen. When indicated, labeled SΔTM spikes were incubated with 300 nM soluble hACE2 in PBS for 90 min before imaging; the hACE2 concentration was maintained during imaging. In addition, a final concentration of 0.5% (w/v) 3-[(3-cholamidopropyl) dimethylammonio]-1-propanesulfonate (CHAPS; Sigma-Aldrich, St. Louis, MO, USA) or 0.01% (w/v) n-Dodecyl β-D-maltoside (DDM; Sigma-Aldrich, St. Louis, MO, USA) was included in the imaging buffers when indicated.

smFRET data were collected using Micromanager[39] v2.0 (micromanager.org) at 25 frames/s. All smFRET data were processed and analyzed using the SPARTAN software package (https://www.scottcblanchardlab.com/software) in Matlab (Mathworks, Natick, MA)[40]. smFRET traces were identified according to criteria previously described[28]. In brief, traces that met these criteria were inspected and verified manually: (1) both donor and acceptor trajectories show a single photobleaching event; (2) FRET lasts for a minimal time window of 15 frames before the photobleaching event; (3) the correlation coefficient of donor and acceptor signals is less than 0.1; (4) the signal-to-noise ratio of the total fluorescence signal is greater than 8. Traces from each of the three technical replicates were then compiled into FRET histograms, and the mean probability per histogram bin ± standard error were calculated. Traces were idealized to a three-state HMM (two non-zero-FRET states and a zero-FRET state) using the maximum point likelihood (MPL) algorithm[41] implemented in SPARTAN. The three-state model was selected by comparing the Akaike information criterion (AIC) across multiple different models with a range of state numbers and topologies as previously described[28]. For each model, the maximized log-likelihood was estimated using the MPL algorithm. The AIC values were then calculated using $AIC_i = 2 \cdot N_i - 2 \cdot LL_i$, in which $N_i$ and $LL_i$ are the number of model parameters and the maximized log-likelihood per trace for the $i$th model. The idealizations were used to determine the occupancies (fraction of time until photobleaching) in each FRET state, and construct Gaussian distributions of each FRET state, which were overlaid on the FRET histograms to visualize the results of the HMM analysis.

## Reporting summary
Further information on research design is available in the Nature Portfolio Reporting Summary linked to this article.

## Data availability

The cryo-EM density maps determined at different detergent conditions have been deposited in the Electron Microscopy Data Bank (EMDB) with the accession codes as follows: EMD-29428, EMD-29430, EMD-29431, EMD-29432, EMD-29434, EMD-29435, EMD-29436, EMD-29438, EMD-29444, EMD-29445, EMD-29446, EMD-29460, EMD-29461, EMD-29462, EMD-29463, EMD-29464, EMD-29465, EMD-29466, EMD-29467, EMD-29468, EMD-29469, EMD-29470, EMD-29471, EMD-29472, EMD-29473, EMD-29474, EMD-29475, EMD-29476, EMD-29477, EMD-29478, EMD-29479, EMD-29525, EMD-29527, EMD-29528, EMD-29529. Raw cryo-EM datasets are available upon request. The single-molecule FRET data generated in this study are provided in the Source Data file. Source data are provided with this paper.

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

## Acknowledgements

We thank all members of the Shen, Munro, and Luban Laboratories for helpful insights, and R. Davis and A. Korostelev for critical reading of the manuscript. This work was supported by grants from the Massachusetts Consortium on Pathogen Readiness (FP-0034266) and the National

Institute of Allergy and Infectious Diseases of the National Institute of Health (R37AI147868 and R01AI148784) to J.L., UMass Chan COVID-19 Pandemic Relief Fund and the National Institute of Health (R01GM143773 and R01AI174645) to J.B.M., and the Worcester Foundation for Biomedical Science, the Massachusetts Consortium on Pathogen Readiness (FP-0034281), the American Heart Association (928304), and the National Institute of General Medical Sciences of the National Institute of Health (R35GM146824) to K.S.

## Author contributions

S.B.E., J.L., and K.S. conceptualized the project. S.B.E. purified the proteins for cryo-EM structural determination. X.W. and N.V.D. collected cryo-EM data. X.W., N.V.D., and S.B.E. analyzed cryo-EM data and built structural models. M.A.D.-S. and J.B.M. design TIRF microscopy experiments. M.A.D.-S. performed TIRF microscopy experiments, including labeling and preparation of proteins. M.A.D.-S. and J.B.M. analyzed TIRF microscopy data. S.B.E., X.W., M.A.D.-S., J.B.M., and K.S. wrote and edited the manuscript with input from all other authors.

## Competing interests

X.W. and N.D. are employees of ThermoFisher Scientific. The remaining authors declare no competing interests.
