## [Peer Review File · Nature Communications]

Detergent modulates the conformational equilibrium of SARS-CoV-2 Spike during cryo-EM structural determinationREVIEWER COMMENTS

Reviewer #1 (Remarks to the Author):

The manuscript by Egri et al. described an effort to identify the impact of detergent on the conformations of SARS-CoV-2 spike protein. The authors found that some detergents triggered SARS-CoV-2 spike protein to take a more closed conformation in cryo-EM settings, but did not affect the conformation of the spike protein in aqueous solutions (as detected by FRET). The study is important in clarifying inconsistent reports in the literature about the conformational states of SARS-CoV-2 spike protein. I have the following comments:

- (1) The manuscript should give a more comprehensive review of the cryo-EM literature to find out the correlation between the used detergent and the observed conformations of the spike protein. Some cryo-EM studies used CHAPS or CHAPS-like detergents (such as CHAPSO) and still observed open spikes. However, these studies were not cited in this manuscript.
- (2) In addition to the use of detergent, another complication of the cryo-EM literature is the use of proline mutations in the S2 subunit. It has been reported that proline mutations can affect the conformations of spike protein. Since the current study didn't introduce any proline mutations, how would the cryo-EM observations in this study be compared with those in other studies that introduced proline mutations (or were carried out under some other different experimental conditions)?
- (3) I am not sure how homogenous the hetero-S trimer was for the FRET assay. Did each hetero-S trimer contain one labeled NTD and one labeled RBD? How can this be confirmed or evaluated? If more than one NTD or RBD is labeled, then the FRET signal can be complicated in situations where more than 1 RBD becomes open.
- (4) Line 213: Why didn't study examine the conformation of D614 spike at neutral pH after it already examined the same protein at low pH? Why use the literature as a comparison here? Please see comment #2.
- (5) Lines 315-318: it is difficult to understand the speculation on the presence of detergent and redistribution of particles in thin vitrified ice. Were the authors suggesting that the detergent somehow changed the physical property of vitrified ice (but not that of aqueous solutions) to make the spike protein more closed? I wonder how this speculation makes physical or chemical sense.
- (6) Figure 1: Protein purification results should go to supplementary data. Also, it would be better to view the conformations of the RBD from the side view (instead of the top view) of the spike protein.
- (7) Figure 2: What is the accuracy of the cryo-EM analysis? Can the cryo-EM analysis be accurate enough to differentiate 10% and 16% closed particles?
Also, why report the particle ratio in both figure labeling and figure legend?
- (8) Figure 3a: I don't think elution volumes of 13.9 ml and 13.8 ml can be used as evidence of conformational changes here. What is the accuracy of the size chromatography experiment?
Again, why report the particle ratio in both figure labeling and figure legend?
- (9) Figure 4: are their error bars for these graphs in Fig. 4C?
- (10) Table 1: is this table redundant with some parts of the figures?

Reviewer #2 (Remarks to the Author):

The manuscript entitled "Detergent modulates the conformational equilibrium of SARS-CoV-2 Spike during cryo-EM structural determination" by Egri, et al. reports a cryo-EM and smFRET study of the effects that detergents and pH have on the conformational dynamics of a SARS-CoV-2 Spike construct comprised of the S1 and S2 subdomains and lacking its transmembrane domains. Specifically, the authors have studied the 'up' and 'down' transitions of the receptor binding domain (RBD) of two variants of this Spike construct, D614 and D614G, that have different up-down dynamics. They have studied these conformational dynamics in the two Spike variants as a function of the identity and concentration of two detergents (CHAPS and DDM) and two pHs (7.4

and 5). Overall, the cryo-EM and smFRET experiments have been carefully executed, and the resulting data are very high quality and have been carefully analyzed and interpreted.

My major concern with this manuscript is that it is highly specialized and, in my opinion, much too specialized for a general audience journal such as Nat Commun. It is a very limited and specific study (the effect of two concentrations of two detergents and two pHs on one conformational change in two variants of one protein). I imagine the results reported here would be of interest to structural biologists studying the SARS-CoV-2 Spike, but I'm not convinced it will have an impact beyond that. It seems more appropriate for a more specialized journal.

A second important concern is the lack of correspondence between some of the state populations inferred from the classification of particles in cryo-EM data and those inferred from the kinetic modeling of smFRET data. I'm not sure what can be made of this. Although there is a lot of excitement about the possibility that state populations inferred from the classification of particles in cryo-EM data can be used to obtain information on the conformational dynamics of biomolecular systems, I don't think the validity of such an approach has yet been demonstrated for any biomolecular system. Consequently, it is not clear to me that one should even expect the state populations inferred from the cryo-EM data to agree with those inferred from the smFRET data, even if the underlying conformational dynamics probed by both techniques were identical. Then, there is the question of whether one should expect that the underlying conformational dynamics probed by the two techniques could even be identical, given the large differences in the experimental conditions and molecular constructs used for cryo-EM and smFRET experiments. One such consideration is the effect that temperature might have on the conformational dynamics probed by cryo-EM. Even though cryo-EM samples are vitrified by very rapid cooling, there is emerging evidence in the literature that biomolecules can conformationally relax during these cooling periods, altering the conformational dynamics that are probed by cryo-EM. A second consideration, this time on the smFRET side, is that it is unknown how the attachment of fluorophores and, when needed, the mutations and peptide tags that are needed to attach the fluorophores to the biomolecular system alter the conformational dynamics that are probed by smFRET. In the current study, the fluorophores and peptide tags are on the larger side of what is typically used for smFRET studies (I think they are fluorophore-quencher conjugates attached to ~8 amino-acid peptide tags that vs. more standard, stand-alone fluorophores linked to single cysteines). I imagine that labels and tags can alter the conformational dynamics of the biomolecular construct of interest and I imagine the larger the labels and tags are, the more likely they are to alter them. These are just two examples. There are various other experimental considerations similar to these that could potentially alter the conformational dynamics observed by cryo-EM and smFRET and could contribute to the discrepancies observed in the current study.

We would first like to thank both reviewers for their time and comments. We believe that the quality of our manuscript has significantly improved after we clarified the points of issue and concern. Our responses are in blue text. All the figure numbers, page numbers, and line numbers below are based on the revised version of the manuscript.

Reviewer #1 (Remarks to the Author):

The manuscript by Egri et al. described an effort to identify the impact of detergent on the conformations of SARS-CoV-2 spike protein. The authors found that some detergents triggered SARS-CoV-2 spike protein to take a more closed conformation in cryo-EM settings, but did not affect the conformation of the spike protein in aqueous solutions (as detected by FRET). The study is important in clarifying inconsistent reports in the literature about the conformational states of SARS-CoV-2 spike protein.

We thank Reviewer 1 for his/her time and comments, and appreciate their acknowledgement of the importance of our study.

I have the following comments:

1. The manuscript should give a more comprehensive review of the cryo-EM literature to find out the correlation between the used detergent and the observed conformations of the spike protein. Some cryo-EM studies used CHAPS or CHAPS-like detergents (such as CHAPSO) and still observed open spikes. However, these studies were not cited in this manuscript.

We agree with Reviewer 1 and have added a more comprehensive summary of the Spike cryo-EM literature (new Supplementary Table 1), that puts the importance of our findings in context.

2. In addition to the use of detergent, another complication of the cryo-EM literature is the use of proline mutations in the S2 subunit. It has been reported that proline mutations can affect the conformations of spike protein. Since the current study didn't introduce any proline mutations, how would the cryo-EM observations in this study be compared with those in other studies that introduced proline mutations (or were carried out under some other different experimental conditions)?

We apologize for the misunderstanding. We had indeed used the stabilizing proline mutations in our Spike constructs. We have updated the manuscript methods section to reflect this important experimental detail. The majority of structural studies on Spike protein uses the stabilizing proline mutations (new Supplementary Table 1), thus our results are broadly applicable and relevant to other Spike structural studies.

3. I am not sure how homogenous the hetero-S trimer was for the FRET assay. Did each hetero-S trimer contain one labeled NTD and one labeled RBD? How can this be confirmed or evaluated? If more than one NTD or RBD is labeled, then the FRET signal can be complicated in situations where more than 1 RBD becomes open.

The untagged S Δ TM construct (D614 or D614G) and tagged S Δ TM plasmid were co-transfected at a 2:1 molar ratio. This approach ensures that, within the purified homotrimer, only one protomer on average is tagged and thus can be visualized specifically. Trimers with more than one tagged protomer are identifiable after data acquisition on the basis of their greater fluorescence intensity and can be disregarded from analysis. This approach has been used in our previous study (Díaz-Salinas *et al.*, eLife, 2022), and we have clarified this in the Methods section.

4. Line 213: Why didn't the study examine the conformation of D614 spike at neutral pH after it already examined the same protein at low pH? Why use the literature as a comparison here? Please see comment #2.

In the very first study of the SARS-CoV-2 Spike protein structure (Walls *et al.*, Cell, 2020), the authors did not use detergent, and our study chose to follow the identical protocol (i.e., Spike construct, protein concentration, and cryo-EM collection parameters). In the Walls *et al.* paper, they evaluated the conformation of D614 Spike at physiological pH under identical conditions, so we can compare our results directly with theirs. We have clarified this point in our manuscript.

5. Lines 315-318: it is difficult to understand the speculation on the presence of detergent and redistribution of particles in thin vitrified ice. Were the authors suggesting that the detergent somehow changed the physical property of vitrified ice (but not that of aqueous solutions) to make the spike protein more closed? I wonder how this speculation makes physical or chemical sense.

We apologize for the confusion. We speculate that detergents may redistribute particles in vitreous ice, away from the air-water interface as previously shown (Weissenberger *et al.*, Nature Methods, 2021), thus leading to the change of Spike conformation. We have clarified this in the text.

6. Figure 1: Protein purification results should go to supplementary data. Also, it would be better to view the conformations of the RBD from the side view (instead of the top view) of the spike protein.

We have updated Figure 1. We have moved protein purification results to Extended Data Figure 1 and have also included a side view of the RBD in Figure 1.

7. Figure 2: What is the accuracy of the cryo-EM analysis? Can the cryo-EM analysis be accurate enough to differentiate 10% and 16% closed particles? Also, why report the particle ratio in both figure labeling and figure legend?

In our cryo-EM data analysis, each individual Spike particle is classified into one of the defined classes, thus the statistics are representative of millions of individual Spike particles, and therefore ensure very high accuracy. Furthermore, during our cryo-EM data analysis we observed a shift of distribution in all four Spike classes – e.g., a decrease in the open conformation while observing a simultaneous increase in the closed conformation. This anticorrelation further strengthens our result.

In addition, in order to ensure the highest degree of accuracy during our cryo-EM analysis, sample preparation, data collection, and data processing were all performed in an identical manner, as outlined in the Methods section. This ensures that any observed differences in conformation distribution are solely due to changes in the detergent conditions.

We have removed the reporting of particle ratio in the figure legend.

8. Figure 3a: I don't think elution volumes of 13.9 ml and 13.8 ml can be used as evidence of conformational changes here. What is the accuracy of the size chromatography experiment? Again, why report the particle ratio in both figure labeling and figure legend?

The comparison of elution volumes of 13.9 mL and 13.8 mL is used to show that lowering pH in gel-filtration buffer does *not* change the general structure of the Spike protein (e.g. cause complex dissociation or protomer unfolding). This is illustrated by similar elution volume and has a nearly identical elution peak shape across conditions. We have clarified this point in the text.

We have removed the reporting of particle ratio in the figure legend.

9. Figure 4: are their error bars for these graphs in Fig. 4C?

The FRET histograms in Figure 4c and 4e include error bars reporting the standard error (horizontal lines in gray). This information is also included in the figure legend.

10. Table 1: is this table redundant with some parts of the figures?

Parts of the table are redundant with some parts of the figures, but we think a separate table that summarizes all the cryo-EM experiments in one location may be more convenient for the readers. We added a note to the table: Part of the data was taken from Figs. 1-3 for comparison purposes.

Reviewer #2 (Remarks to the Author):

The manuscript entitled “Detergent modulates the conformational equilibrium of SARS-CoV-2 Spike during cryo-EM structural determination” by Egri, et al. reports a cryo-EM and smFRET study of the effects that detergents and pH have on the conformational dynamics of a SARS-CoV-2 Spike construct comprised of the S1 and S2 subdomains and lacking its transmembrane domains. Specifically, the authors have studied the ‘up’ and ‘down’ transitions of the receptor binding domain (RBD) of two variants of this Spike construct, D614 and D614G, that have different up-down dynamics. They have studied these conformational dynamics in the two Spike variants as a function of the identity and concentration of two detergents (CHAPS and DDM) and two pHs (7.4 and 5). Overall, the cryo-EM and smFRET experiments have been carefully executed, and the resulting data are very high quality and have been carefully analyzed and interpreted.

We thank Reviewer 2 for his/her time, insightful comments, and appreciation that our experiments have been “carefully executed” and that our data is “very high quality” and “carefully analyzed and interpreted”.

- My major concern with this manuscript is that it is highly specialized and, in my opinion, much too specialized for a general audience journal such as Nat Commun. It is a very limited and specific study (the effect of two concentrations of two detergents and two pHs on one conformational change in two variants of one protein). I imagine the results reported here would be of interest to structural biologists studying the SARS-CoV-2 Spike, but I’m not convinced it will have an impact beyond that. It seems more appropriate for a more specialized journal.

We apologize for not making this clear. Our results are highly applicable to the broader field of biology.

First, for the structural biology field, detergents have been used heuristically to resolve the preferred orientation issue in cryo-EM structural determination throughout hundreds of studies. To date, there have been very few studies that systematically and directly address how the use of commonly employed detergents alter the protein conformation. Therefore, it

is not known whether the observed difference in conformational states under varying conditions truly reflects the properties of the protein, or is merely affected by detergents. Our results may contribute broadly to other protein structures solved by cryo-EM, and thus are valuable for the structural biology community.

Second, for the field of virology, it is common to correlate viral protein conformation of a particular strain with their biological properties, including transmissibility, infectivity, and immune escape. Therefore, an accurate measure of the Spike conformation is vital to draw the correct conclusion. However, as seen in our analyses, detergents actively modulate the Spike conformation, which leads to disagreement in data interpretation across different studies. Our results have clarified these discrepancies so are of importance to the virology field.

Third, for the field of epidemiology, due to the high societal relevance of understanding the SARS-CoV-2 viral transmission and replication life cycle, we believe a careful and thorough examination of the viral Spike protein conformation to be of high interest and value to the broader scientific community (e.g., researchers using structures to guide antibody-based therapeutics against SARS-CoV-2 Spike protein).

We have emphasized these points in the text to approach a broad audience.

- A second important concern is the lack of correspondence between some of the state populations inferred from the classification of particles in cryo-EM data and those inferred from the kinetic modeling of smFRET data. I'm not sure what can be made of this. Although there is a lot of excitement about the possibility that state populations inferred from the classification of particles in cryo-EM data can be used to obtain information on the conformational dynamics of biomolecular systems, I don't think the validity of such an approach has yet been demonstrated for any biomolecular system. Consequently, it is not clear to me that one should even expect the state populations inferred from the cryo-EM data to agree with those inferred from the smFRET data, even if the underlying conformational dynamics probed by both techniques were identical. Then, there is the question of whether one should expect that the underlying conformational dynamics probed by the two techniques could even be identical, given the large differences in the experimental conditions and molecular constructs used for cryo-EM and smFRET experiments. One such consideration is the effect that temperature might have on the conformational dynamics probed by cryo-EM. Even though cryo-EM samples are vitrified by very rapid cooling, there is emerging evidence in the literature that biomolecules can conformationally relax during these cooling periods, altering the conformational dynamics that are probed by cryo-EM. A second consideration, this time on the smFRET side, is that it is unknown how the attachment of fluorophores and, when

needed, the mutations and peptide tags that are needed to attach the fluorophores to the biomolecular system alter the conformational dynamics that are probed by smFRET. In the current study, the fluorophores and peptide tags are on the larger side of what is typically used for smFRET studies (I think they are fluorophore-quencher conjugates attached to ~8 amino-acid peptide tags that vs. more standard, stand-alone fluorophores linked to single cysteines). I imagine that labels and tags can alter the conformational dynamics of the biomolecular construct of interest and I imagine the larger the labels and tags are, the more likely they are to alter them. These are just two examples. There are various other experimental considerations similar to these that could potentially alter the conformational dynamics observed by cryo-EM and smFRET and could contribute to the discrepancies observed in the current study.

We thank the reviewer for his/her time and insightful comment. Although there is currently a lack of coordination between cryo-EM and smFRET in the study of Spike proteins, we are confident that in the future, these two approaches will be in concert to fully depict the biophysical nature of the Spike protein.

In other fields of study, for example, the ribosome field, cryo-EM and smFRET have been used in tandem to investigate and characterize numerous aspects of translation dynamics. As one example, the large-scale intersubunit rotations that occur between small and large ribosomal subunits has been described first by cryo-EM (Frank and Agrawal, *Nature*, 2000), then confirmed by bulk FRET in solution (Ermolenko *et al.*, *J. Mol. Biol.*, 2007), and lastly the kinetics of rotation were validated using smFRET (Cornish *et al.*, *Mol. Cell.*, 2008). More recently, smFRET was used to guide the identification of intermediate ribosome configurations during tRNA-mRNA translocation, which were then resolved by cryo-EM (Rundlet *et al.*, *Nature*, 2021). Importantly, in all cases, FRET–single-molecule and bulk–and cryo-EM converged on the same mechanistic insight of intersubunit rotation during translocation.

In a perfect world, a real-time atomic movie of the protein would capture all conformational states and dynamics by which the protein interconverts between its conformational states. Such a technique would fully describe all structural transitions between conformational states and would pave the way for a detailed, mechanistic understanding of how the protein functions. However, in the absence of such a technique, cryo-EM can be used to determine structural snapshots of distinct conformational states the protein exists in.

Here, we use cryo-EM to explore four such states of the Spike protein – fully closed, one-RBD up, two-RBD up, and fully open – and assess how the presence of commonly used detergents shifts the distribution of these conformations. Due to the intrinsic nature of cryo-

EM we are unable to capture the transitions of RBD conformational changes. However, this is where smFRET serves as an orthogonal approach to what we observed using cryo-EM. Using a previously validated smFRET setup (Díaz-Salinas *et al.*, eLife, 2022) we were able to monitor the dynamics of RBD conformational changes in real-time. We measured changes in RBD conformation that were correlated with the changes we observed using cryo-EM.

Although the correspondence between the two methods is lacking at this moment, especially given the vastly different experimental differences intrinsic to the experimental methods highlighted by the reviewer, by having two orthogonal approaches that converge on the same conclusion, we can have further confidence and strengthen our finding and understanding of the Spike RBD dynamics. Again, we would like to emphasize that similar correlations across multiple biophysical approaches have been made for a great number of other biological systems and we do not believe that the Spike RBD dynamics are so unique such that using these two biophysical methods in tandem does not apply to our study.

We have modified our discussion section of our manuscript to emphasize these points.

REVIEWERS' COMMENTS

Reviewer #1 (Remarks to the Author):

The authors have addressed my previous comments well.

Reviewer #3 (Remarks to the Author):

This is a timely and well-executed study that presents cryo-EM and smFRET data of high quality.

While I can understand the potential concern raised by reviewer 2 that this study might be perceived as somewhat specialized, it is certainly true that detergents are frequently included during cryo-EM sample preparation to mitigate issues such as the preferred orientation of a specimen. I agree with the authors, however, that a better understanding of the effects of such detergents on protein conformation is urgently needed. From that perspective, this study is certainly of interest to the field of structural biology in general, and to the field of structural virology in particular.

Reviewer 2 also raised the concern that some of the state populations inferred from cryo-EM particle classification may not perfectly match those inferred from the kinetic modeling of smFRET data. However, the authors' cryo-EM and smFRET data are in overall very good agreement. There are now numerous successful examples where cryo-EM and smFRET analyses have been synergistically applied with great success (for a recent review, see Bacic et al, Curr Opin Struct Biol. 2020).

Response to reviewers' comments

Reviewer #1 (Remarks to the Author):

The authors have addressed my previous comments well.

We appreciate the support from Reviewer 1.

Reviewer #3 (Remarks to the Author):

This is a timely and well-executed study that presents cryo-EM and smFRET data of high quality.

While I can understand the potential concern raised by reviewer 2 that this study might be perceived as somewhat specialized, it is certainly true that detergents are frequently included during cryo-EM sample preparation to mitigate issues such as the preferred orientation of a specimen. I agree with the authors, however, that a better understanding of the effects of such detergents on protein conformation is urgently needed. From that perspective, this study is certainly of interest to the field of structural biology in general, and to the field of structural virology in particular.

Reviewer 2 also raised the concern that some of the state populations inferred from cryo-EM particle classification may not perfectly match those inferred from the kinetic modeling of smFRET data. However, the authors' cryo-EM and smFRET data are in overall very good agreement. There are now numerous successful examples where cryo-EM and smFRET analyses have been synergistically applied with great success (for a recent review, see Bacic et al, Curr Opin Struct Biol. 2020).

We thank Reviewer 3 for the positive remarks and appreciate his/her support. We have included the mentioned review in our manuscript.